# A Cancer-Related microRNA Signature Shows Biomarker Utility in Multiple Myeloma

**DOI:** 10.3390/ijms222313144

**Published:** 2021-12-05

**Authors:** Aristea-Maria Papanota, Paraskevi Karousi, Christos K. Kontos, Pinelopi I. Artemaki, Christine-Ivy Liacos, Maria-Alexandra Papadimitriou, Tina Bagratuni, Evangelos Eleutherakis-Papaiakovou, Panagiotis Malandrakis, Ioannis Ntanasis-Stathopoulos, Maria Gavriatopoulou, Efstathios Kastritis, Margaritis Avgeris, Meletios-Athanasios Dimopoulos, Andreas Scorilas, Evangelos Terpos

**Affiliations:** 1Department of Clinical Therapeutics, School of Medicine, National and Kapodistrian University of Athens, 11528 Athens, Greece; ampapanota@med.uoa.gr (A.-M.P.); liakou@med.uoa.gr (C.-I.L.); tbagratuni@med.uoa.gr (T.B.); evelepapa@med.uoa.gr (E.E.-P.); panosmalan@med.uoa.gr (P.M.); johnntanasis@med.uoa.gr (I.N.-S.); mgavria@med.uoa.gr (M.G.); ekastritis@med.uoa.gr (E.K.); mdimop@med.uoa.gr (M.-A.D.); 2Department of Biochemistry and Molecular Biology, Faculty of Biology, National and Kapodistrian University of Athens, 15701 Athens, Greece; pkarousi@biol.uoa.gr (P.K.); chkontos@biol.uoa.gr (C.K.K.); partemaki@biol.uoa.gr (P.I.A.); maria-p@biol.uoa.gr (M.-A.P.); mavgeris@med.uoa.gr (M.A.); 3Laboratory of Clinical Biochemistry-Molecular Diagnostics, Second Department of Pediatrics, School of Medicine, National and Kapodistrian University of Athens, “P. & A. Kyriakou” Children’s Hospital, 11527 Athens, Greece

**Keywords:** miRNA, small non-coding RNA (sncRNA), plasma cell dyscrasias, multiple myeloma bone disease (MMBD), post-transcriptional regulation, target mRNA, KEGG pathways, molecular biomarkers, prognosis, overall survival

## Abstract

Multiple myeloma (MM) is the second most common hematological malignancy, arising from terminally differentiated B cells, namely plasma cells. miRNAs are small non-coding RNAs that participate in the post-transcriptional regulation of gene expression. In this study, we investigated the role of nine miRNAs in MM. CD138+ plasma cells were selected from bone marrow aspirates from MM and smoldering MM (sMM) patients. Total RNA was extracted and in vitro polyadenylated. Next, first-strand cDNA synthesis was performed using an oligo-dT–adapter primer. For the relative quantification of the investigated miRNAs, an in-house real-time quantitative PCR (qPCR) assay was developed. A functional in silico analysis of the miRNAs was also performed. miR-16-5p and miR-155-5p expression was significantly lower in the CD138+ plasma cells of MM patients than in those of sMM patients. Furthermore, lower levels of miR-15a-5p, miR-16-5p, and miR-222-3p were observed in the CD138+ plasma cells of MM patients with osteolytic bone lesions, compared to those without. miR-125b-5p was also overexpressed in the CD138+ plasma cells of MM patients with bone disease that presented with skeletal-related events (SREs). Furthermore, lower levels of miR-223-3p were associated with significantly worse overall survival in MM patients. In conclusion, we propose a miRNA signature with putative clinical utility in MM.

## 1. Introduction

Multiple myeloma (MM) is a neoplasm arising from plasma cells. It accounts for 1–1.8% of all malignancies and is the second commonest hematological malignancy [1]. MM develops in a multistep process, starting from its asymptomatic precursor states: monoclonal gammopathy of undetermined significance (MGUS) and smoldering multiple myeloma (sMM). All these disease entities are part of a spectrum of diseases referred to as plasma cell dyscrasias. The major difference between MM and its precursors is that MM is characterized by end-organ damage, triggered by the monoclonal protein and/or the cytokines secreted by malignant plasma cells. The clinical manifestations of the disease are hypercalcemia, anemia, renal impairment, and bone disease [2].

MM pathogenesis is complex, and the exact mechanism of MM progression remains unknown. MM arises from plasma cells as a result of genetic alterations such as chromosomal defects and epigenetic alterations. Chromosomal translocations are the primary genetic events leading to MM and premalignant states preceding MM. Secondary genetic events, such as acquired mutations, are more frequent while progressing from MGUS and sMM to MM. The interaction between malignant plasma cells and the bone marrow microenvironment also plays a key role in MM pathogenesis and progression [3]. The bone marrow microenvironment consists of a variety of hematopoietic cells, such as T cells, B cells, and NK cells, as well as osteoclasts and other non-hematopoietic cells such as osteoblasts, stromal cells, and endothelial cells. Factors secreted by those cells and myeloma cells also play an important role in cell-to-cell interactions, leading to disease progression, drug resistance, and bone destruction. In the bone marrow microenvironment, exosomes carrying microRNAs (miRNAs) play an important role in cell-to-cell communication [4].

miRNAs are regulatory, small non-coding RNA molecules that control gene expression at the post-transcriptional level. They are transcribed as primary transcripts (pri-miRNAs) and undergo multistep enzymatic processing to become mature miRNA molecules [5]. Mature miRNA molecules are loaded in Argonaute proteins and form the RNA-induced silencing complex (RISC), which mediates RNA interference. The miRNA guides RISC to a specific target mRNA [6]. miRNAs feature a “seed” sequence that recognizes and binds to a region of homology found most frequently in the 3′ untranslated region (UTR) of the target mRNAs, but also in the 5′ UTR or within the coding region. Once the binding has occurred, the translation procedure is interrupted and/or the target mRNA is degraded [7]. A single miRNA can have multiple targets; moreover, some cellular pathways are regulated by single miRNAs or miRNAs deriving from clusters [8]. miRNAs can regulate a variety of biological functions and their deregulation leads to human disease, particularly to malignancy [9]. miRNAs seem to play an important role in multiple myeloma by regulating the expression of oncogenes and tumor suppressor genes, as well as by regulating the bone marrow microenvironment, leading to disease progression, drug resistance, and bone destruction [10,11,12,13].

In this study, we aimed to investigate the potential clinical utility of a group of cancer-related miRNAs, namely miR-15a-5p, miR-16-5p, miR-21-5p, miR-25-3p, miR-125b-5p, miR-155-5p, miR-221-3p, miR-222-3p, and miR-223-3p, as biomarkers in MM. For this purpose, we developed real-time quantitative PCR (qPCR) assays to quantify their levels in the bone marrow CD138+ plasma cells of MM and sMM patients. Moreover, a functional in silico analysis was performed to investigate the putative role(s) of the clinically significant miRNAs in MM.

## 2. Results

### 2.1. Development and Optimization of Real-Time qPCR Assays

In order to quantify each miRNA in all the samples, we developed and optimized qPCR assays. After the standardization of the primer concentration, thermal protocol, and quantity of cDNA input, a unique melting curve of each investigated miRNA and of the reference genes was observed, as shown in Figure 1.

### 2.2. miR-16-5p and miR-155-5p Levels Are Significantly Lower in MM Patients Compared to sMM Patients

We observed a statistically significant reduction in the levels of miR-16-5p and miR-155-5p in CD138+ plasma cells of MM patients compared to those of sMM patients, using the Mann–Whitney *U* test (*p* = 0.036 and 0.045 respectively) (Figure 2). The miR-16-5p levels in the MM CD138+ plasma cells were equal to a mean ± standard error (S.E.) of 13.77 ± 4.10 RQU, while in the sMM CD138+ plasma cells, the average levels of this miRNA were 44.58 ± 14.04 RQU. Regarding miR-155-5p, a mean ± S.E. of 366.0 ± 167.6 RQU was observed in the MM patients’ CD138+ plasma cells and a mean ± S.E. of 1285.7 ± 539.8 RQU in the sMM patients’ CD138+ plasma cells. These results are shown in Table 1.

### 2.3. Association of the Expression of Three of the Investigated miRNAs with MM Bone Disease (MMBD)

Using the Mann–Whitney *U* test, we observed significantly lower levels of miR-15a-5p (*p* = 0.037), miR-16-5p (*p* = 0.035), and miR-222-3p (*p* = 0.037) in the CD138+ cells of MM patients with osteolytic bone lesions, compared to those of MM patients without. These results are illustrated in Figure 3. Regarding miR-15a-5p, we observed a mean ± S.E. of 7.71 ± 3.25 RQU in patients with osteolytic lesions and a mean ± S.E. of 29.93 ± 12.66 RQU in patients without. The mean levels of miR-16-5p were equal to 7.49 ± 2.39 RQU in those MM patients presenting with osteolysis, and 19.27 ± 7.18 RQU in patients without osteolysis. Last, miR-222-3p levels in MM patients with osteolytic lesions in WBLDCT were 6.03 ± 1.65 RQU, compared to 10.95 ± 3.17 RQU in patients without osteolytic bone disease. These results are shown in Table 1.

### 2.4. miR-125b-5p Levels Are Associated with MMBD Severity

In the subgroup of patients with MMBD, the levels of miR-125b-5p were significantly higher in MM patients presenting SREs (mean ± S.E. = 39.39 ± 12.72 RQU) compared to those without SREs (mean ± S.E. = 15.00 ± 8.12 RQU) (*p* = 0.005). These results are presented in Figure 3D and Table 1.

### 2.5. miR-223-3p Offers a Putative Prognostic Value in MM

Having determined an optimal cut-off point for prognostic purposes, we sub-grouped the MM patients, based on miR-223-3p levels into two distinct groups, namely miR-223-3p-positive and miR-223-3p-negative. This cut-off point was 22.19 RQU, equal to the 45th percentile. A univariate bootstrap Cox regression analysis showed that MM patients with lower levels of miR-223-3p in their CD138+ cells show significantly shorter OS intervals (Hazard ratio (HR): 3.11, bootstrap *p* = 0.034). A multivariate bootstrap Cox regression analysis showed that low miR-223-3p levels retain their adverse prognostic significance independently of the Revised International Staging System (R-ISS) (HR: 3.34, bootstrap *p* = 0.046). These results are presented in Table 2. Kaplan–Meier survival analysis curves verified these results, as lower levels of miR-223-3p were shown to be related to significantly shorter OS intervals (*p* = 0.046) (Figure 4).

### 2.6. In Silico Functional miRNA Analysis

The following miRNAs were included in the functional in silico analysis: miR-15a-5p, miR-16-5p, miR-125b-5p, miR-155-5p, miR-222-3p, and miR-223-3p. These miRNAs were grouped as follows: miR-16-5p, miR-155-5p, and miR-223-3p formed the first group, as they were associated with disease progression; miR-15a-5p, miR-16-5p, miR-125b-5p, and miR-222-3p formed the second group, as they were associated with bone disease. Forty-two significantly enriched KEGG pathways were obtained for the first group, among which ubiquitin-mediated proteolysis, the TNF signaling pathway, the TGF-β signaling pathway, protein processing in the endoplasmic reticulum, the P53 signaling pathway, the Hippo signaling pathway, the cell cycle, and pathways in cancer, as well as many solid and hematological malignancies were included. In the second group, thirty-five (35) significantly enriched KEGG pathways were obtained, among which ubiquitin-mediated proteolysis, the TNF signaling pathway, the TGF-β signaling pathway, proteoglycans in cancer, the Hippo signaling pathway, RNA transport, and adherence junctions were included. These results are shown in Figure 5.

## 3. Discussion

As mentioned above, miRNAs are regulatory molecules implicated in the vast majority of biological procedures. Their deregulation is associated with human diseases, including cancer [9]. Within this prism, scientific research is currently guided towards the investigation of their role in the pathophysiology of malignancy, as well as their putative role as diagnostic and prognostic biomarkers [14]. The role of miRNAs and other small non-coding RNAs in multiple myeloma and its associated bone disease is a current topic of scientific interest [13,15,16,17]. In this study, we investigated the role of a group of miRNAs with an established role in human malignancy in MM and proved their association with the clinicopathological characteristics and the disease outcome of MM patients.

This is not the first time that miR-16-5p has been investigated in MM patients. The *MIR16* and *MIR15A* genes are located in chromosome 13q [18]. Given the fact that del(13q) is a frequent chromosomal abnormality in MM, and considering that this deletion is associated with MM patients’ prognosis, researchers extensively investigated the role of those two miRNAs in MM [19]. Their studies revealed that these two miRNAs are downregulated in MM cases, compared to normal controls [20,21,22]. miR-15a-5p and miR-16-5p inhibit cell proliferation in MM by targeting *VEGF* and *CABIN1*, as well as inhibiting the AKT and NF-κB pathways [23], all of which have been shown to play an important role in the MM molecular background. Therefore, the downregulation of miR-16-5p and miR-15a-5p in MM leads to disease promotion, through the upregulation of their direct targets [22,24]. When miR-16-5p and miR-15a-5p levels were assessed in serum samples from MM and its precursor states, sMM and MGUS, in comparison with healthy donors, the levels of both miRNAs were found to be lower in patients with plasma cell dyscrasias than in normal controls [25]. More specifically, circulating miR-16-5p levels were shown to gradually decrease, while progressing from a normal state to MM. In this study, we also observed lower levels of miR-16-5p in CD138+ plasma cells of MM patients compared to CD138+ plasma cells of sMM patients, indicating that not only the circulating but also the intracellular miR-16-5p levels decrease during MM progression. In the same context, miR-155-5p expression was found to be lower in MM patients’ plasma cells, compared to plasma cells of healthy controls, in several studies [26,27]. As we showed that lower miR-155-5p levels are found in CD138+ plasma cells of MM patients compared to those of sMM patients, it is possible that there is a gradual reduction in miR-155-5p levels during the multistep process of myelomagenesis.

To the best of our knowledge, this is the first time that miR-15a-5p, miR-16-5p, miR-222-3p, and miR-125b-5p are shown to be associated with MMBD. Although all these molecules were extensively investigated in MM [28,29,30] and associated with osteogenesis and other forms of bone disease, such as osteoporosis [31,32,33,34,35,36], no association with MMBD has been described so far. Currently, a variety of diagnostic modalities are available for the diagnosis of MMBD. WBLDCT is the new gold standard for the diagnosis of MMBD, while whole-body magnetic resonance imaging (MRI) and positron emission tomography-computed tomography (PET-CT) are also available for specific cases, in which the assessment of diffuse infiltration or viability of focal lesions is needed [37]. Nevertheless, despite the remarkable evolution of imaging modalities, no established blood biomarkers able to support the diagnosis of MMBD or indicate its severity are available. These biomarkers would be of great importance in ambiguous cases, where setting a diagnosis in order to guide therapeutic decisions is difficult. It would be interesting to investigate the utility of the miRNAs shown to be associated with the presence of osteolytic lesions and SREs as diagnostic biomarkers of MMBD in large cohorts of MM patients presenting with and without MMBD. In the era of denosumab, targeted treatments are the new focus in the management of MMBD. These molecules could serve as possible targets for precision therapy in MMBD. Currently, treatments for MMBD inhibit osteoclastic activity, but there is still an unmet need for medicines that augment osteoblastic activity in MM. Given that some of these miRNAs are implicated in the differentiation of mesenchymal stem cells towards osteoblasts, they could serve as candidate target molecules for such treatment.

The results of this study highlight the putative prognostic value of miR-223-3p in MM. More specifically, low expression levels of this miRNA were predictive of inferior OS, and its prognostic significance was independent of the R-ISS. The R-ISS staging system is currently the most reliable tool used to predict survival in MM. There are three prognostic subgroups with different survival intervals, namely R-ISS I, R-ISS II, and R-ISS III [38]. R-ISS stage II is the largest and most heterogeneous class, constituting 62% of the total MM population. There is an unmet need for molecular biomarkers able to further classify patients with R-ISS stage II in terms of their prognosis. miR-223-3p could serve as a prognostic biomarker able to predict prognosis in these patients. For this purpose, large, randomized trials are needed to evaluate the prognostic significance of this molecule in large MM cohorts.

Functional in silico analysis in two distinct miRNA groups was performed. Among the enriched pathways for the miRNAs associated with disease progression, several were found to play a critical role in MM. For instance, the in silico analysis showed that the investigated miRNAs are implicated in the regulation of ubiquitin-mediated proteolysis, while protein processing in the endoplasmic reticulum (ER) was also found to be regulated by those miRNAs. ER detects misfolded proteins and promotes their ubiquitination and degradation at the proteasome. In MM, large quantities of immunoglobulins are produced by plasma cells, leading to an increased ER protein burden. Therefore, ER is stressed, and this stress leads to the activation of signaling pathways that reduce protein production. Proteasome inhibitors (PIs) constitute a drug category of great value regarding MM treatment. PIs block the proteasomal degradation of the misfolded proteins causing elevated ER stress, resulting in the apoptosis of malignant plasma cells [39]. Hence, it would be interesting to investigate the role of this miRNA group in response to therapy. The ubiquitin-mediated proteolysis pathway was also enriched in the group of miRNAs associated with MMBD; this pathway seems to have an important role in MMBD due to the reported role of E3 ubiquitin ligase in the regulation of osteoblast differentiation and bone formation [40].

Additionally, many signaling pathways associated with MM progression were found to be enriched. The P53 signaling pathway constitutes such an example, as its deregulation also plays an important role in the development and prognosis of MM. In newly diagnosed patients with MM, three types of P53 deregulation were observed; the deletion of *TP53* (del(17p)), monoallelic mutation, and biallelic inactivation. Del(17p) is a well-established prognostic factor in MM related to poor prognosis and is included in the high-risk cytogenetics of the R-ISS staging system [38]. The biallelic inactivation of *P53* is also related to unfavorable disease outcomes, while the role of monoallelic mutations needs to be elucidated [41].

Interestingly, some signaling pathways were found to be enriched in both analyses concerning the miRNAs associated with disease progression and miRNAs associated with MMBD. The TNF signaling pathway represents such a pathway. TNF is an important cytokine that acts in the bone marrow microenvironment, inducing plasma cell differentiation and MM development [42]. High levels of TNF seem to be associated with aggressive disease, while particular TNF polymorphisms are associated with an increased risk of MM development [43,44]. Additionally, RANK (TNFRSF11A) and OPG (TNFRSF11B) receptors are members of the TNF superfamily, and the balance between RANKL (TNFSF11) and OPG regulates the osteoclast-induced bone resorption [45]. According to this functional in silico analysis, the TGF-β signaling pathway was also enriched in both miRNA groups. In MM, TGF-β is secreted in the bone marrow microenvironment by both MM cells and bone marrow stromal cells, resulting in the increased secretion of VEGF and IL6 by stromal cells, thereby leading to the proliferation of malignant cells [46]. Moreover, TGF-β is implicated in MMBD both by inhibiting osteoblast formation and inducing osteoclast maturation [47]. Another enriched pathway in both miRNA groups is the Hippo pathway; in MM, the YAP1 protein, which is part of the Hippo pathway, acts as a tumor suppressor, as the rescue of YAP1 has been shown to promote ABL1-dependent apoptosis in hematological malignancies. Malignant plasma cells express STK4 and SIRT6 that act as YAP1 inhibitors. As a result, apoptosis is inhibited and MM cell survival is promoted [48,49]. Moreover, YAP1 protein binds to β-catenin (also known as CTNNB1), thus augmenting the pre-osteogenic role of β-catenin [50] and proving the role of the pathway in terms of MMBD development. Within this prism, TAZ (also known as TAFAZZIN) protein binds and activates RUNX2, which is a transcription factor with an established role in osteoblastogenesis [51]. Malignant plasma cells secrete TNF and FGF2 that both inhibit the TAZ protein [52]. According to scientific data, bortezomib, which was the first PI approved for MM treatment, inhibits FGF2 and,, as a result, restores TAZ levels, leading to increased osteoblast formation [53].

RNA transport, a function that was found to be enriched in the functional in silico analysis of miRNAs associated with MMBD, is also a significant function, since circulating miRNAs that are transported in extracellular vesicles (EVs) seem to play an important role in MMBD development. Many studies indicate miRNAs as cargo of MM-derived EVs and, more specifically, several studies proved the presence of MMBD-related miRNAs in MM-derived EVs [54]. Lastly, proteoglycans in cancer was another significantly enriched pathway for miRNAs associated with MMBD. As mentioned above, the balance between RANKL and OPG regulates osteoclast-induced bone resorption; MM cells produce a heparin sulfate proteoglycan named Syndecan-1 (CD138). This proteoglycan binds to OPG and induces its degradation causing as a result activation of the RANK/RANKL pathway, increased osteoclast formation, and MMBD [55]. All these data, accompanied by the functional in silico analysis results, indicate that miRNAs could serve as promising therapeutic targets for MMBD, while this field merits investigation.

Our study features some limitations. Firstly, the cohort size could be larger, but it is important to underline that the patient cohort is highly representative of the real-world MM population, since no age or performance status exclusion criteria were followed. Moreover, the median follow-up time is 2 years. A longer follow-up period could possibly reveal stronger associations between the investigated molecules and MM patients’ survival.

In summary, our study highlighted the clinical significance of a group of miRNAs in MM. We provide evidence that miR-16-5p and miR-155-5p merit further investigation as diagnostic biomarkers, able to discriminate between MM and sMM. Moreover, to the best of our knowledge, we describe, for the first time, an association between miR-15a-5p, miR-16-5p, miR-222-3p, and miR-125b-5p with MMBD. We also show that miR-223-3p is likely to offer favorable prognostic value in MM. Finally, functional in silico analysis of the investigated miRNAs revealed their implication in a variety of signaling pathways that are of great significance in MM.

## 4. Materials and Methods

### 4.1. Study Participants

Bone marrow aspirate (BMA) samples were collected, at the time of diagnosis, from ninety-four (94) adult patients, newly diagnosed with MM (76 patients) or sMM (18 patients) at the Department of Clinical Therapeutics of the “Alexandra” General Hospital of Athens. The patients who had already been subjected to any type of anti-myeloma therapy or suffered from any other concomitant malignancy were excluded from this study. The assigned physicians provided the participants with concise information about the procedures, the methods, and the aims of the study. Afterwards, written informed consent was obtained from each participant. This study was also approved by the scientific board of the “Alexandra” General Hospital of Athens and was conducted according to the principles of the Declaration of Helsinki. Following informed consent acquisition, the BMA samples were collected.

The clinical characteristics of the MM patients are described in Table 3. The median age of the MM patients was 68 years (range: 35–88 years). Only one patient did not receive any treatment, whereas the treatment of the other 75 patients varied. In more detail, 63 out of the 75 MM patients (84.0%) were treated with bortezomib plus an immunomodulatory drug (either lenalidomide (60 cases) or thalidomide (3 cases)), 6 (8.0%) MM patients received bortezomib combined with cyclophosphamide and dexamethasone, and 6 (8.0%) MM patients were treated with lenalidomide in combination with dexamethasone. Moreover, 21 (27.6%) out of 76 MM patients were subjected to autologous stem cell transplantation following high-dose melphalan, while the remaining 55 (72.4%) MM patients were ineligible for bone marrow transplant, either because they were older than 65 years (48 cases) or because of severe comorbidities and/or impaired performance status (7 cases). Bone disease was assessed via whole-body low-dose computed tomography (WBLDCT). Skeletal-related events (SREs) of MM patients presenting with osteolysis are also presented in Table 3.

### 4.2. CD138+ Plasma Cell Selection

Ten (10) mL of BMA from each participant were collected in tubes containing ethylenediaminetetraacetic acid (EDTA). The BMA samples were processed immediately after collection for CD138+ selection. The BMA mononuclear cells were first isolated using the Ficoll–Paque technique. After that, anti-CD138–coated magnetic beads (Miltenyi Biotech, Bergisch Gladbach, Germany) were used to perform a positive selection of CD138+ plasma cells.

### 4.3. RNA Isolation, In Vitro Polyadenylation, and Reverse Transcription

TRI Reagent^®^ (Molecular Research Center Inc., Cincinnati, OH, USA) was used to isolate the total RNA from CD138+ plasma cells. A Qubit™ 2 Fluorometer (Invitrogen™, Thermo Fisher Scientific Inc., Carlsbad, CA, USA) was used to measure the RNA concentration of the extracts. Following that, 200 ng of each total RNA extract were in vitro polyadenylated using *E. coli* Poly(A) Polymerase (New England Biolabs Ltd., Ontario, ON, Canada) and 80 μM ATP, at 37 °C for 60 min, followed by an inactivation step at 65 °C for 10 min. Subsequently, the in vitro polyadenylated RNA samples were reversely transcribed using MMLV reverse transcriptase (Invitrogen™, Thermo Fisher Scientific Inc., Carlsbad, CA, USA) and an oligo-dT–adapter primer, as previously described [56].

### 4.4. Quantification of miRNA Expression Using Real-Time qPCR

In order to quantify the expression of miR-15a-5p, miR-16-5p, miR-21-5p, miR-25-3p, miR-125b-5p, miR-155-5p, miR-221-3p, miR-222-3p, and miR-223-3p, we developed and optimized the respective SYBR Green-based qPCR assays. The sequences and other features of the primers used in the qPCR assays are shown in Appendix A. Due to the short length of the miRNA sequence, the specificity in both SYBR Green and TaqMan assays is determined by the whole length of the miRNA sequence; thus, an optimized SYBR Green assay-based can be as accurate as a TaqMan probe-based assay [57]. For our qPCR assays, a specific forward primer was designed for each miRNA as well as for *SNORD43* and *SNORD48*, which were used as reference genes. A universal reverse primer, complementary to the oligo-dT–adapter primer used during cDNA synthesis (Appendix A), was used, as previously described [58]. The reaction mixture was composed using 0.5 μL of 10-fold diluted cDNA, 5 μL KAPA™ SYBR^®^ FAST qPCR master mix (2X) (Kapa Biosystems Inc., Woburn, MA, USA), 1 μL of each primer at a final concentration of 200 nM each, and 2.5 μL RNase-free H_2_O. The qPCR was performed in an ABI 7500 Fast Real-Time PCR System (Applied Biosystems™, Thermo Fisher Scientific Inc., Foster City, CA, USA).

The levels of each miRNA were determined using the comparative C_T_ (2^−ΔΔCt^) method [59]. We measured the levels of each miRNA molecule in relative quantification units (RQUs), as the ratio of each miRNA to the geometric mean of *SNORD48* and *SNORD43* molecules, divided by the same ratio determined for a cDNA pool, consisting of 5 sample cDNAs, used as a calibrator. The equations describing the aforementioned calculations are the following:(1)$${\Delta \mathrm{Ct}}_{\mathrm{miRNA}\left(\mathrm{sample}\right)}={\mathrm{Ct}}_{\mathrm{miRNA}\left(\mathrm{sample}\right)}-\frac{{\mathrm{Ct}}_{SNORD43\left(\mathrm{sample}\right)}+{\mathrm{Ct}}_{SNORD48\left(\mathrm{sample}\right)}}{2}$$
(2)$${\Delta \Delta \mathrm{Ct}}_{\mathrm{miRNA}\left(\mathrm{sample}\right)}={\Delta \mathrm{Ct}}_{\mathrm{miRNA}\left(\mathrm{sample}\right)}-{\Delta \mathrm{Ct}}_{\mathrm{miRNA}\left(\mathrm{calibrator}\right)}$$
(3)$${\mathrm{RQ}}_{\mathrm{miRNA}\left(\mathrm{sample}\right)}=2^{-\Delta \Delta \mathrm{Ct}\left(\mathrm{miRNA},\mathrm{sample}\right)}$$

### 4.5. Biostatistics

The data were extensively analyzed using SPSS^®^ software (version 26) (IBM Corporation, Armonk, NY, USA). Due to the non-Gaussian distribution of the expression of each miRNA in both cohorts of patients, the Mann–Whitney *U* test was performed to determine the statistical significance of the differences observed in miRNA levels between the MM and sMM patients, as well as between subgroups of MM patients.

Moreover, overall survival (OS) and progression-free survival (PFS) analyses for the cohort of MM patients were performed. For this purpose, the X-tile algorithm was applied to generate the optimal prognostic cut-off points [60]. Kaplan–Meier OS and PFS curves were built, and differences between them were assessed using the Mantel–Cox (log-rank) test. Next, univariate and multivariate Cox regression analyses were performed. Bootstrapping (1000 samples) was applied to estimate the bias-corrected and accelerated (BCa) 95% confidence intervals (CI) of each hazard ratio (HR).

For each statistical test that was performed, the results were considered statistically significant only when the *p* value was <0.050.

### 4.6. Functional In Silico Analysis for miRNA Target Prediction and KEGG Pathway Analysis

Those miRNAs shown to feature putative biomarker utility in MM were subjected to functional in silico analysis. Thus, two groups of miRNAs were generated; the first included those miRNAs that were shown to be related to disease progression, while the second consisted of those miRNAs that seemed to be related to bone disease. DIANA miRPath v.3.0 [61] was used to predict putative miRNA targets, before proceeding to the Kyoto Encyclopedia of Genes and Genomes (KEGG) pathway analysis. Tarbase v.7.0 was used to detect the putative miRNA targets and a *p* value threshold at 0.050 was applied. Last, “Pathways union” option was used to integrate the results. In order to visualize the results, the R programming language was used; specifically, ggplot2 was applied to create graphics, along with “dplyr”, “hrbrthemes”, and “viridis” R packages.

## Figures and Tables

**Figure 1 ijms-22-13144-f001:**
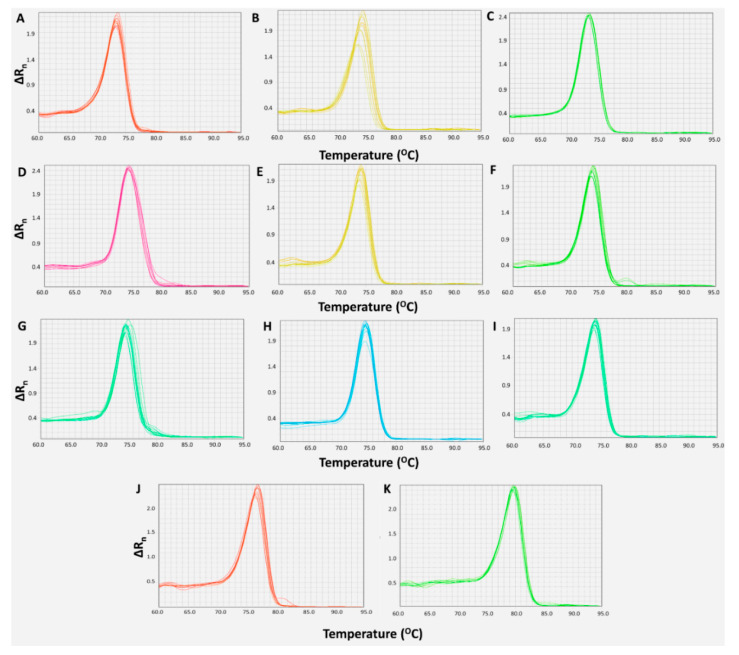
Melting curves of miR-15a-5p (**A**), miR-16-5p (**B**), miR-21-5p (**C**), miR-25-3p (**D**), miR-125b-5p (**E**), miR-155-5p (**F**), miR-221-3p (**G**), miR-222-3p (**H**), miR-223-3p (**I**), *SNORD43* (**J**), and *SNORD48* (**K**), generated after their amplification under the optimized conditions of the real-time quantitative PCR (qPCR) assays. The graphs were built by plotting ΔRn, which represents the difference in the fluorescence from the baseline, against temperature.

**Figure 2 ijms-22-13144-f002:**
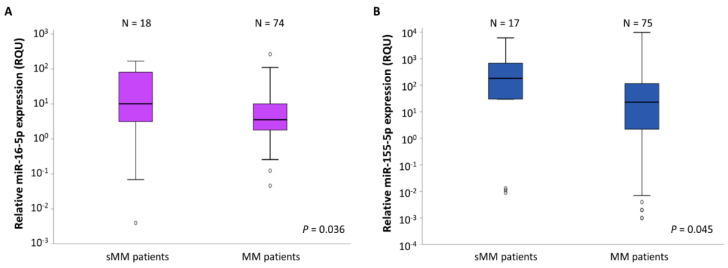
Boxplots showing the relative expression of miR-16-5p (**A**) and miR-155-5p (**B**) in CD138+ plasma cells of smoldering multiple myeloma (sMM) and multiple myeloma (MM) patients. The bold line inside each box represents the median value, while the lower and upper borders of the box represent the 25th (Q1) and 75th (Q3) percentiles, respectively; the area within the box represents the interquartile range (IQR). The lower and upper whiskers show values equal to Q1−1.5∗IQR and Q3+1.5∗IQR, respectively. The circles outside the boxes indicate outliers. The scale of the vertical axis is logarithmic.

**Figure 3 ijms-22-13144-f003:**
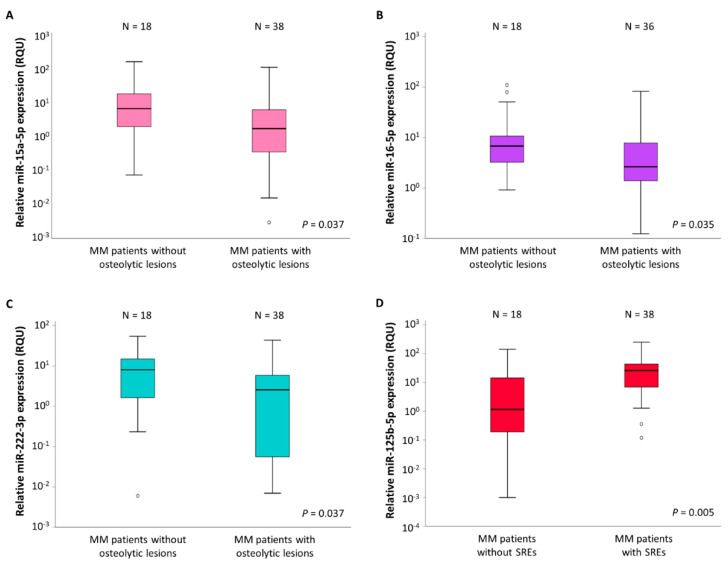
Boxplots showing the relative expression of miR-15a-5p (**A**), miR-16-5p (**B**), and miR-222-3p (**C**) in CD138+ plasma cells of multiple myeloma (MM) patients without and with osteolytic lesions. (**D**) Boxplot showing the relative expression of miR-125b-5p in CD138+ plasma cells of MM patients without and with skeletal-related events (SREs). The bold line inside each box represents the median value, while the lower and upper borders of the box represent the 25th (Q1) and 75th (Q3) percentiles, respectively; the area within the box represents the interquartile range (IQR). The lower and upper whiskers show values equal to Q1−1.5∗IQR and Q3+1.5∗IQR, respectively. The circles outside boxes indicate outliers. The scale of the vertical axis is logarithmic.

**Figure 4 ijms-22-13144-f004:**
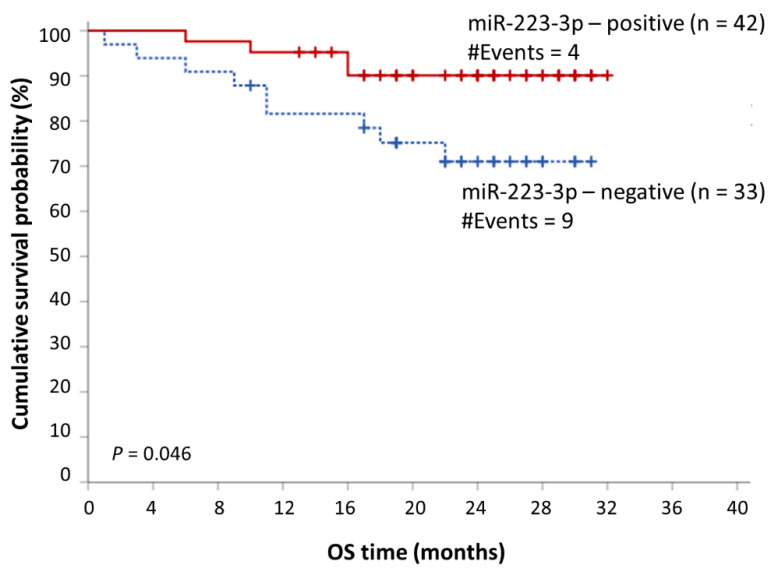
Kaplan–Meier OS curves showing the prognostic potential of miR-223-3p expression in CD138+ plasma cells of MM patients.

**Figure 5 ijms-22-13144-f005:**
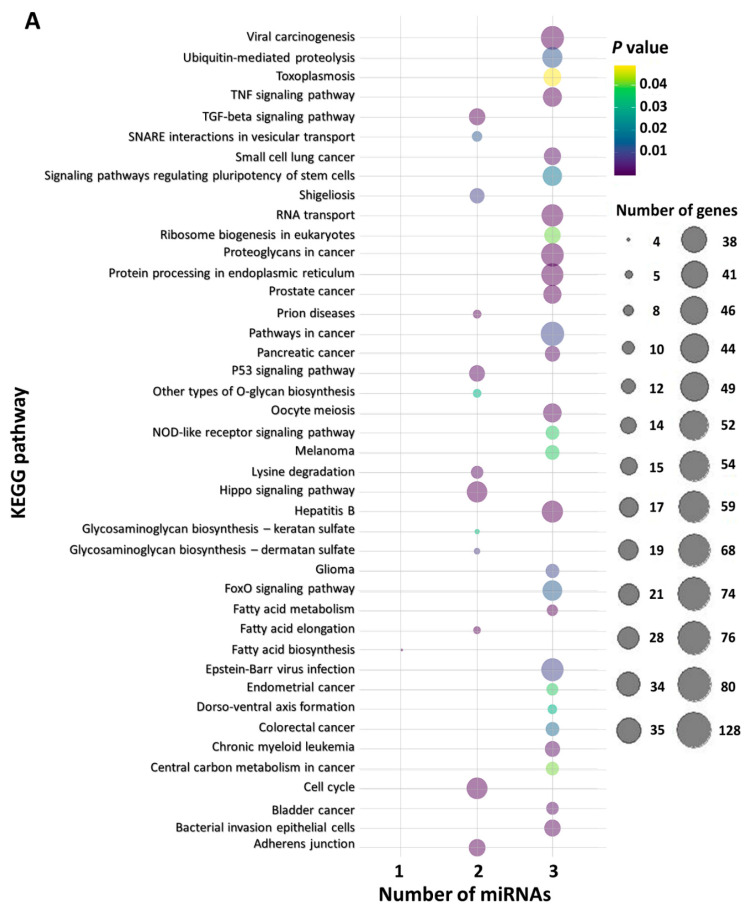
The KEGG pathways in which miR-16-5p, miR-155-5p, and miR-223-3p (**A**) as well as miR-15a-5p, miR-16-5p, miR-125b-5p, and miR-222-3p (**B**) possibly participate. The size of each bubble shows the number of miRNA targets related to each pathway, while the color of the bubble indicates the likelihood of the implication in each pathway.

**Table 1 ijms-22-13144-t001:** Distribution of miRNA expression in CD138+ plasma cells of patients with smoldering MM and MM, as well as in CD138+ plasma cells of MM patient subgroups.

Variable	Mean ± S.E. ^1^	Range	Percentiles		
			25th	50th (Median)	75th
**miR-16-5p levels (RQU ^2^)**					
in sMM patients	44.58 ± 14.04	0.004–167.3	2.94	10.16	82.20
in MM patients	13.77 ± 4.10	0.046–264.5	1.77	3.54	10.26
**miR-155-5p levels (RQU ^2^)**					
in sMM patients	1285.7± 539.8	<0.001–6097.5	15.06	182.8	1402.7
in MM patients	366.0 ± 167.6	<0.001–9738.0	2.21	23.05	130.3
**miR-15a-5p levels (RQU ^2^)**					
in MM patients without osteolytic lesions	29.93 ± 12.66	0.008–181.0	1.76	7.30	22,29
in MM patients with osteolytic lesions	7.71 ± 3.25	0.003–122.9	0.37	1.85	6.76
**miR-16-5p levels (RQU ^2^)**					
in MM patients without osteolytic lesions	19.27 ± 7.18	0.92–109.9	3.07	7.06	15.68
in MM patients with osteolytic lesions	7.49 ± 2.39	0.12–82.77	1.39	2.64	8.50
**miR-222-3p levels (RQU ^2^)**					
in MM patients without osteolytic lesions	10.95 ± 3.17	0.006–54.28	1.56	8.03	15.57
in MM patients with osteolytic lesions	6.03 ± 1.65	0.007–43.54	0.06	2.56	6.55
**miR-125b-5p levels (RQU ^2^)**					
in MM patients without SREs ^3^	15.00 ± 8.12	0.001–139.8	0.17	1.17	15.62
in MM patients with SREs ^3^	39.39 ± 12.72	0.12–247.7	5.96	26.28	44.75

^1^ Standard error. ^2^ Relative quantification unit. ^3^ Skeletal-related events.

**Table 2 ijms-22-13144-t002:** Univariate and multivariate Cox regression analyses, regarding MM patients’ overall survival (OS).

	Univariate Analysis			Multivariate Analysis		
Covariate	HR ^1^	BCa ^4^ Bootstrap ^5^ 95% CI ^2^	Bootstrap^5^ *p* Value ^3^	HR ^1^	BCa ^4^ Bootstrap ^5^ 95% CI ^2^	Bootstrap^5^ *p* Value ^3^
**miR-223-3p expression status**						
Positive	1.00			1.00		
Negative	3.11	0.95–22.15	*0.034*	3.34	0.71–2.6 × 10^5^	*0.046*
**R-ISS ^6^ (ordinal)**	3.31	1.05–13.22	*0.025*	3.14	1.05–21.17	*0.021*

^1^ Hazard ratio. ^2^ Confidence interval. ^3^ Italics indicate a significant *p* value. ^4^ Bias-corrected and accelerated. ^5^ Based on 1000 bootstrap samples. ^6^ Revised International Staging System.

**Table 3 ijms-22-13144-t003:** Characteristics of the 76 multiple myeloma (MM) cases.

Variable	Number of MM Patients (%)
**Gender**	
Male	44 (57.9%)
Female	32 (42.1%)
**Myeloma type**	
IgG	44 (57.9%)
IgA	17 (22.4%)
IgD	2 (2.6%)
Light chain	10 (13.2%)
Non-secretory	2 (2.6%)
Missing data	1 (1.3%)
**ISS ^1^ stage**	
I	15 (19.7%)
II	25 (32.9%)
III	34 (44.8%)
Missing data	2 (2.6%)
**R-ISS ^2^ stage**	
I	11 (14.5%)
II	40 (52.6%)
III	18 (23.7%)
Missing data	7 (9.2%)
**Bone disease**	
No	22 (28.9%)
Yes	50 (65.8%)
Missing data	4 (5.3%)
**WBLDCT ^3^ osteolysis**	
No	18 (23.7%)
Yes	38 (50.0%)
Missing data	20 (26.3%)
**SREs ^4^ (38 MM patients)**	
No	18 (47.4%)
Yes	20 (52.6%)

^1^ International Staging System. ^2^ Revised International Staging System. ^3^ Whole-body low-dose computed tomography. ^4^ Skeletal-related events; only patients with osteolysis are assessed for the presence of SREs.

## Data Availability

The data presented in this study are available on request from the corresponding author. The data are not publicly available due to ethical issues.

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
