# Peer review of "A Cancer-Related microRNA Signature Shows Biomarker Utility in Multiple Myeloma"

_ijms, 2021, doi:10.3390/ijms222313144_

Round 1

Reviewer 1 Report

The manuscript presented by Papanota et al, entitled “A Cancer-related microRNA Signature Shows Biomarker Util-2 ity in Multiple Myeloma” investigates the role of a group of miRNAs in Multiple myeloma (MM).

The authors present data, showing that miR-16-5p and miR-155-5p levels are significantly lower in MM patients compared to smoldering MM (sMM) patients. Also, they found that the lower levels of miR-15a-5p, miR-16-5p, and 30 miR-222-3p were observed in MM patients with osteolytic bone lesions compared to those without. Moreover, another miRNA they investigated, miR-125b-5p was overexpressed in MM patients with bone disease that presented with skeletal-related events (SREs). Furthermore, lower levels of miR-223-3p were associated with significantly shorter overall survival intervals.

In general, the material is well organized and it is easy to be followed and understood.

The single remark that can improve the manuscript is enlisted below:

Even that the figures and the tables are quite informative, their legends could be expanded, so they would become more informative.  

In my opinion, the current version of the manuscript is suitable for publishing after minor corrections.

Author Response

The manuscript presented by Papanota et al, entitled “A Cancer-related microRNA Signature Shows Biomarker Util-2 ity in Multiple Myeloma” investigates the role of a group of miRNAs in Multiple myeloma (MM).

The authors present data, showing that miR-16-5p and miR-155-5p levels are significantly lower in MM patients compared to smoldering MM (sMM) patients. Also, they found that the lower levels of miR-15a-5p, miR-16-5p, and 30 miR-222-3p were observed in MM patients with osteolytic bone lesions compared to those without. Moreover, another miRNA they investigated, miR-125b-5p was overexpressed in MM patients with bone disease that presented with skeletal-related events (SREs). Furthermore, lower levels of miR-223-3p were associated with significantly shorter overall survival intervals.

In general, the material is well organized and it is easy to be followed and understood.

 The single remark that can improve the manuscript is enlisted below:

Even that the figures and the tables are quite informative, their legends could be expanded, so they would become more informative. 

In my opinion, the current version of the manuscript is suitable for publishing after minor corrections.

We thank the Reviewer for his positive appraisal of our original research article. In response to his remark, we made the Figure legends more explanatory.

  • Page 6, lines 220-223: Figure 1. Melting curves of miR-15a-5p (A), miR-16-5p (B), miR-21-5p (C), miR-25-3p (D), miR-125b-5p (E), miR-155-5p (F), miR-221-3p (G), miR-222-3p (H), miR-223-3p (I), SNORD43 (J), and SNORD48 (K), generated after their amplification under optimized conditions of the real-time quantitative PCR (qPCR) assays.
  • Page 6, lines 225-231: Figure 2. Boxplots showing the relative expression of miR-16-5p (A) and miR-155-5p (B) in CD138+ plasma cells of smoldering multiple myeloma (sMM) and multiple myeloma (MM) patients. The bold line inside each box represents the median value, while the lower and upper borders of the box represent the 25th (Q1) and 75th (Q3) percentiles, respectively; the area within the box represents the interquartile range (IQR). The lower and upper whiskers show values equal to Q1-1.5*IQR and Q3+1.5*IQR, respectively. The circles outside boxes indicate outliers. The scale of the vertical axis is logarithmic.

  • Page 8, lines 253-260: Figure 3. Boxplots showing the relative expression of miR-15a-5p (A), miR-16-5p (B), and miR-222-3p (C) in CD138+ plasma cells of multiple myeloma (MM) patients without and with osteolytic lesions. (D) Boxplot showing the relative expression of miR-125b-5p in CD138+ plasma cells of MM patients without and with skeletal-related events (SREs). The bold line inside each box represents the median value, while the lower and upper borders of the box represent the 25th (Q1) and 75th (Q3) percentiles, respectively; the area within the box represents the interquartile range (IQR). The lower and upper whiskers show values equal to Q1-1.5*IQR and Q3+1.5*IQR, respectively. The circles outside boxes indicate outliers. The scale of the vertical axis is logarithmic.
  • Page 9, lines 266-267: Figure 4. Kaplan-Meier OS curves, showing the prognostic potential of miR-223-3p expression in CD138+ plasma cells of MM patients.
  • Page 11, lines 286-289: Figure 5. The KEGG pathways in which miR-16-5p, miR-155-5p, and miR-223-3p (A) as well as miR-15a-5p, miR-16-5p, miR-125b-5p, and miR-222-3p (B) possibly participate. The size of each bubble shows the number of miRNA targets related to each pathway, while the color of the bubble indicates the likelihood of the implication in each pathway.

The Authors wish to thank the Reviewers for their constructive comments that led to the improvement of the current manuscript.

Reviewer 2 Report

The manuscript by Papanota et al. describes microRNAs, which were previously associated with MM disease, to be associated with bone disease in MM patients. They isolated CD138+ cells from bone marrow aspirates and from these isolated microRNAs, whose levels are quantified by SBYR green. Although the manuscript is interesting and brings some novelty, several major points need to be clarified:

Methods section

Line 140-143: The authors state that: The levels of each miRNA were determined using the comparative threshold cycle (CT) (2−ΔΔCt) method [17]. Further, they express the levels in relative quantification units (RQUs). It is however not clear to me how was it determined? Was then the 2−ΔΔCt value expressed as a ratio? Please use rather an equation, to present the calculation, as it is very unclear, also with the ratios of calibrators…

Why authors used SYBR green as a method of detection of microRNAs and not generally accepted and validated TaqMan probes?

Results section

Line 179-183: The authors state that they observed a statistically significant reduction in the levels of miR-16-5p and miR-155-5p in plasma cells from MM patients compared to those from sMM patients, using the Mann-Whitney U test. Further, they describe mean ± SEM values for each miRNA in each cohort. However, these mean values and SEM values are not recapitulated by the figure 2, where it is not indicated if the boxplots represent mean or median and if mean, it does not resemble these data. For example: miR-16-5p levels in MM plasma cells = 13.77 ± 4.1 RQU, while in sMM = 44.58 ± 14.01 RQU. However, in the figure 2, mean miR-16-5p levels for MM are around 5 and for sMM are around 10. Please clarify and in the Figure 2 legend indicate what do the boxplots represent.

The same applies for a section about miRNAs miR-15a-5p (P=0.037), miR-16-5p (P=0.035), and miR-222-3p and Figure 3, where proper legend is missing and the described data are not recapitulated by the figures. Moreover, the same applies for miR-125b-5p levels and Figure 3D, which does not recapitulate the described results.

Section 3.5 miR-223-3p has a putative prognostic value in MM

It is not clear at which cut-off where the patients divide to have high or low expression of miR-223-3p. Is the level of miR-223-3p different between MM and sMM patients?

 Moreover, Figure 4 shows very immature data, where only 4 vs 9 events are presented in high vs low expressing miRNA cohorts. These data should be taken with caution and may be misleading at the moment. Instead, is there any relationship with progression free survival in MM patients and levels of miR-223-3p?

Section 3.6 Functional miRNA analysis. It is not clear to me, why all miRNAs were tested here at once. In this analysis only miR-16-5p and miR-155-5p were different between MM and sMM, whereas miR-15a-5p, miR-16-5p and miR-222-3p were different in MM patients with or without bone disease. Thus, KEGG enriched pathways should be presented separately for miRNAs which are associated with disease progression or with bone disease formation. At the same time, the authors should discuss the KEGG enriched pathways in regards to bone disease formation, not only to pathogenesis or to prognosis of MM.

Author Response

  1. Line 140-143: The authors state that: The levels of each miRNA were determined using the comparative threshold cycle (CT) (2−ΔΔCt) method [17]. Further, they express the levels in relative quantification units (RQUs). It is however not clear to me how was it determined? Was then the 2−ΔΔCt value expressed as a ratio? Please use rather an equation, to present the calculation, as it is very unclear, also with the ratios of calibrators…

We thank the Reviewer for this comment. We added the appropriate equations to make this clear:

Page 4, lines 158-163:

The equations describing the aforementioned calculations are the following:

  1. Why authors used SYBR green as a method of detection of microRNAs and not generally accepted and validated TaqMan probes?

As miRNAs are small molecules (≈22 nucleotides, the specificity in both SYBR Green and Taq-Man assays is determined by the whole length of the miRNA sequence, and therefore our in-house-developed optimized SYBR green assay is as specific as a Taq-Man assay. Additionally, studies have proved that when optimizing the SYBR Green method, its performance and quality could be comparable to TaqMan method.

We clarified this in our manuscript:

Page 4, lines 142-145: Due to the small length of miRNA sequence, the specificity in both SYBR Green and Taq-Man assays is determined by the whole length of the miRNA sequence; thus, an opti-mized SYBR Green assay-based can be as accurate as a TaqMan probe-based assay [15].

We also added an appropriate reference:

  1. Tajadini, M.; Panjehpour, M.; Javanmard, S.H. Comparison of SYBR Green and TaqMan methods in quantitative real-time polymerase chain reaction analysis of four adenosine receptor subtypes. Adv Biomed Res 2014, 3, 85, doi:10.4103/2277-9175.127998

  1. Line 179-183: The authors state that they observed a statistically significant reduction in the levels of miR-16-5p and miR-155-5p in plasma cells from MM patients compared to those from sMM patients, using the Mann-Whitney U test. Further, they describe mean ± SEM values for each miRNA in each cohort. However, these mean values and SEM values are not recapitulated by the figure 2, where it is not indicated if the boxplots represent mean or median and if mean, it does not resemble these data. For example: miR-16-5p levels in MM plasma cells = 13.77 ± 4.1 RQU, while in sMM = 44.58 ± 14.01 RQU. However, in the figure 2, mean miR-16-5p levels for MM are around 5 and for sMM are around 10. Please clarify and in the Figure 2 legend indicate what do the boxplots represent. The same applies for a section about miRNAs miR-15a-5p (P=0.037), miR-16-5p (P=0.035), and miR-222-3p and Figure 3, where proper legend is missing and the described data are not recapitulated by the figures. Moreover, the same applies for miR-125b-5p levels and Figure 3D, which does not recapitulate the described results.

In response to the Reviewer’s remark, we explained that the median values are represented in both Figures 2 and 3. Moreover, we explained that the scale of the vertical axis is logarithmic in all boxplots. Additionally, we added a Table showing the distributions of miRNA levels, including median values.

  • Page 7, lines 225-231: Figure 2. Boxplots showing the relative expression of miR-16-5p (A) and miR-155-5p (B) in CD138+ plasma cells of smoldering multiple myeloma (sMM) and multiple myeloma (MM) patients. The bold line inside each box represents the median value, while the lower and upper borders of the box represent the 25th (Q1) and 75th (Q3) percentiles, respectively; the area within the box represents the interquartile range (IQR). The lower and upper whiskers show values equal to Q1-1.5*IQR and Q3+1.5*IQR, respectively. The circles outside boxes indicate outliers. The scale of the vertical axis is logarithmic.
  • Page 7, lines 232-233: Table 2. Distribution of miRNA expression in CD138+ plasma cells of patients with smoldering MM and MM, as well as in CD138+ plasma cells of MM patients’ subgroups.
  • Page 8, lines 253-260: Figure 3. Boxplots showing the relative expression of miR-15a-5p (A), miR-16-5p (B), and miR-222-3p (C) in CD138+ plasma cells of multiple myeloma (MM) patients without and with osteolytic lesions. (D) Boxplot showing the relative expression of miR-125b-5p in CD138+ plasma cells of MM patients without and with skeletal-related events (SREs). The bold line inside each box represents the median value, while the lower and upper borders of the box represent the 25th (Q1) and 75th (Q3) percentiles, respectively; the area within the box represents the interquartile range (IQR). The lower and upper whiskers show values equal to Q1-1.5*IQR and Q3+1.5*IQR, respectively. The circles outside boxes indicate outliers. The scale of the vertical axis is logarithmic.

  1. It is not clear at which cut-off where the patients divide to have high or low expression of miR-223-3p. Is the level of miR-223-3p different between MM and sMM patients?

We thank the Reviewer for this remark. Apparently, miR-223-3p patients do not differ between sMM and MM patients, but within distinct groups of MM patients, stratified according to miR-223-3p expression. We clarified this in the Results section.

Page 7, lines 241-244: After having determined an optimal cut-off point for prognostic purposes, we sub-grouped MM patients based on miR-223-3p levels into two distinct groups, namely miR-223-3p-positive and miR-223-3p-negative. This cut-off point was 22.19 RQU, equal to the 45th percentile.

  1. Moreover, Figure 4 shows very immature data, where only 4 vs 9 events are presented in high vs low expressing miRNA cohorts. These data should be taken with caution and may be misleading at the moment. Instead, is there any relationship with progression free survival in MM patients and levels of miR-223-3p?

We agree with the Reviewer, that these results should be taken with caution. Therefore, we performed bootstrap univariate Cox regression analysis, using 1000 random samples, to enhance our results. Moreover, no relationship was observed between progression-free survival in MM patients and levels of miR-223-3p.

Page 7, lines 241-243: Univariate bootstrap Cox regression analysis showed that MM patients with lower levels of miR-223-3p in their CD138+ cells show significantly shorter OS intervals (Hazard ratio (HR): 3.11, bootstrap P=0.034).

Additionally, we discuss this as a limitation of our study:

Page 15, lines 478-483: Our study is subjected to some limitations. Firstly, the size of our cohort could be larger, but at that point, it is important to underline that our population is highly suggestive of the real-world MM population, since no age or performance status exclusion criteria were followed. Moreover, the median follow-up time is 2 years. A longer follow-up period of our cohort could possibly reveal stronger associations between the investigated molecules and patients’ survival.

  1. Section 3.6 Functional miRNA analysis. It is not clear to me, why all miRNAs were tested here at once. In this analysis only miR-16-5p and miR-155-5p were different between MM and sMM, whereas miR-15a-5p, miR-16-5p and miR-222-3p were different in MM patients with or without bone disease. Thus, KEGG enriched pathways should be presented separately for miRNAs which are associated with disease progression or with bone disease formation. At the same time, the authors should discuss the KEGG enriched pathways in regards to bone disease formation, not only to pathogenesis or to prognosis of MM.

We agree with this Reviewer’s remark. We performed the functional analyses separately, for two miRNA groups, based on their associations with either MM progression or bone disease.

Page 5, lines 180-183: Those miRNAs shown to have a putative biomarker utility in MM were subjected to functional in silico analysis. Thus, two groups of miRNAs were generated; the first one included those miRNAs that were shown to be related to disease progression, while the second one consisted of those miRNAs that seemed to be related to bone disease.

Moreover, we added one new Figure, which shows these results separately:

Page 11, lines 286-289: Figure 5. The KEGG pathways in which miR-16-5p, miR-155-5p, and miR-223-3p (A) as well as miR-15a-5p, miR-16-5p, miR-125b-5p, and miR-222-3p (B) possibly participate. The size of each bubble shows the number of miRNA targets related to each pathway, while the color of the bubble indicates the likelihood of the implication in each pathway.

Last, we modified a large part of the Discussion, to comment on these Results separately:

Pages 13-15, lines 378-477: Functional in silico analysis in two distinct miRNA groups was performed. Forty-two (42) KEGG pathways were significantly enriched for the first miRNA group that consisted of the miRNAs associated with disease progression (miR-16-5p, miR-155-5p, and miR-223-3p). Among the enriched pathways, many of them were found to play a critical role in malignancies. The P53 signaling pathway constitutes such an example. More specifically, P53 is a well-known tumor suppressor, and somatic mutations in the gene encoding P53 are described in more than 50% of the human malignancies [50]. Moreover, its deregulation also plays an important role in the development and prognosis of MM. In newly diagnosed patients with MM, three types of P53 deregulation have been observed; deletion of TP53 [del(17p)], monoallelic mutation, and biallelic inactivation. Del(17p) is a well-established prognostic factor in MM related to poor prognosis and is included in the high-risk cytogenetics of the R-ISS staging system [49]. Biallelic inactivation of P53 is also related to unfavorable disease outcomes, while the role of monoallelic mutations needs to be elucidated [51].

Moreover, our study indicated that the investigated miRNAs are implicated in the regulation of ubiquitin-mediated proteolysis. The ubiquitin-proteasome pathway regulates the degradation of damaged or misfolded proteins or proteins that are no longer useful for cellular functions [52]. Proteasome inhibitors (PIs) constitute a drug category of great value regarding MM treatment. Bortezomib was the first PI approved for MM treatment, followed by carfilzomib and ixazomib [3]. PIs lead MM cells to apoptosis. This could be achieved, among other mechanisms, via activation of P53 signaling pathway [53]. Furthermore, protein processing in the endoplasmic reticulum (ER) was found to be regulated by those miRNAs. ER detects misfolded proteins and promotes their ubiquitination and degradation at the proteasome. In MM, large quantities of immunoglobulins are produced by plasma cells leading to increased ER protein burden. Therefore, ER is stressed and this leads to the activation of signaling pathways that reduce protein pro-duction. PIs block proteasomal degradation of the misfolded proteins causing elevated ER stress, resulting in the apoptosis of the malignant plasma cells [54]. Hence, it would be in-teresting to investigate the role of this miRNA group in response to therapy.

Another signaling pathway, which was also enriched in the functional in silico anal-ysis of the first miRNA group investigated and plays a key role in MM, is the TNF signaling pathway. TNF is an important cytokine that acts in the bone marrow microenvironment inducing plasma cell differentiation and MM development [55]. High levels of TNF seem to be associated with aggressive disease, while particular TNF polymorphisms are associated with an increased risk of MM development [56,57]. According to this functional in silico analysis, the TGF-β signaling pathway is also regulated by the aforementioned miRNAs. The TGF-β signaling pathway is normally implicated in cellular functions, including differentiation, proliferation, and survival. During normal hematopoiesis, this pathway induces differentiation and negatively regulates proliferation [58]. In MM, TGF-β is secreted in the bone marrow microenvironment by both MM cells and bone marrow stromal cells leading to increased secretion of VEGF and IL6 by stromal cells leading to the proliferation of the malignant cells [59]. Another enriched pathway is the Hippo pathway, a recently discovered pathway implicated in human malignancies [60]. In MM the YAP1 protein, which is part of the Hippo pathway, acts as a tumor suppressor, as rescue of YAP1 has been shown to promote ABL1-dependent apoptosis in hematological malignancies. Malignant plasma cells express STK4 and SIRT6 that act as YAP1 inhibitors. As a result, apoptosis is inhibited and MM cell survival is promoted [61,62]. Therefore, these investigated miRNAs could add a regulatory layer in these pathways and in their implication in MM development and progression.

Thirty-five (35) KEGG pathways were significantly enriched for the second miRNA group that consisted of the miRNAs associated with MMBD (miR-15a-5p, miR-16-5p, miR-125b-5p, and miR-222-3p). Among them, several molecular pathways with an important role in MMBD were enriched. Ubiquitin-mediated proteolysis seems to have an important role in MMBD due to the reported role of E3 ubiquitin ligase in the regulation of osteoblast differentiation and bone formation [63]. Interestingly, the TNF signaling pathway is regulated by the majority of the miRNAs of the second group. The RANK/RANKL pathway is one of the most important pathways implicated in MMBD and RANK (TNFRSF11A) is a member of the TNF superfamily. More specifically, RANK is expressed on the surface of premature osteoclasts. The binding of RANKL (TNFSF11) to RANK is crucial for osteoclast maturation. Osteoprotegerin (OPG; also known as TNFRSF11B) is another TNF family member that binds to RANKL thus preventing its binding to the RANK receptor causing as a final result the inhibition of osteoclast maturation [64]. In MM, the balance between RANKL and OPG is disrupted, favoring RANKL and leading to increased osteoclast-induced bone resorption [65].

The TGF-β signaling pathway, which was also enriched, is implicated in MMBD both by inhibiting osteoblast formation and inducing osteoclast maturation. Activin A be-longs to the TGF-β superfamily and promotes osteoclast maturation mainly through a SMAD-dependent pathway that results in RANK secretion and activation of the NF-κB pathway [66]. Activin A is elevated in the body fluids of MMBD patients and high levels of circulating activin Α are associated with adverse disease outcomes [67]. Bone morphogenetic proteins (BMPs) also belong to the TGF-β superfamily. BMP2 promotes osteoblastogenesis [68]. In MM malignant plasma cells overexpress inhibitors of BMP2 induced osteoblastogenesis, such as hepatocyte growth factor (HGF) [69]. The Hippo signaling pathway was also enriched in the second group of miRNAs in our analysis. This path-way, which was discussed above under the prism of its role in MM progression, is also implicated in MMBD [70]. The Hippo pathway has an important role in osteoblast maturation. TAZ (also known as TAFAZZIN) protein binds and activates RUNX2, which is a transcription factor with an established role in osteoblastogenesis [71]. Malignant plasma cells secrete TNF and FGF2 that both inhibit the TAZ protein [72]. According to scientific data, bortezomib inhibits FGF2 and as a result, restores TAZ levels leading to increased osteoblast formation [73]. Moreover, YAP1 protein binds to β-catenin (also known as CTNNB1) thus augmenting the pre-osteogenic role of β-catenin [74].

RNA transport, a function found enriched in the functional in silico analysis, is also a significant function regarding MMBD, since circulating miRNAs that are transported in extracellular vesicles (EVs) seem to have an important role in MMBD development. Many studies indicate miRNAs as cargo of MM-derived EVs and more specifically several studies proved the presence of MMBD related miRNAs in MM-derived EVs [75]. For instance, miR-129-5p is a miRNA that blocks osteogenic differentiation by targeting and downregulating SPI1 a transcription factor regulating pre-osteogenic gene expression. Researchers proved that miR-129-5p expression was higher in EVs derived from MM patients com-pared to those from sMM patients, and treatment of mesenchymal stem cells with EVs from MM patients lead to increased miR-129-5p expression in mesenchymal stem cells and abrogation of osteogenesis [76]. Proteoglycans in cancer was another significantly enriched pathway. As we discussed above, OPG acts as a decoy receptor for RANKL and inhibits its binding to RANK thus blocking osteoclast maturation. MM cells produce a heparin sulfate proteoglycan named syndecan-1. This proteoglycan binds to OPG and induces its degradation causing as a result activation of the RANK/RANKL pathway, increased osteoclast formation and MMBD [77]. All these data, accompanied by the functional in silico analysis results, indicate that miRNAs could serve as promising therapeutic targets for MMBD, while this field merits investigation.

We also added the appropriate references:

  1. Perri, F.; Pisconti, S.; Della Vittoria Scarpati, G. P53 mutations and cancer: a tight linkage. Ann Transl Med 2016, 4, 522, doi:10.21037/atm.2016.12.40.
  2. Flynt, E.; Bisht, K.; Sridharan, V.; Ortiz, M.; Towfic, F.; Thakurta, A. Prognosis, Biology, and Targeting of TP53 Dysregulation in Multiple Myeloma. Cells 2020, 9, doi:10.3390/cells9020287.
  3. King, R.W.; Deshaies, R.J.; Peters, J.M.; Kirschner, M.W. How proteolysis drives the cell cycle. Science 1996, 274, 1652-1659, doi:10.1126/science.274.5293.1652.
  4. Nunes, A.T.; Annunziata, C.M. Proteasome inhibitors: structure and function. Seminars in oncology 2017, 44, 377-380, doi:10.1053/j.seminoncol.2018.01.004.
  5. Obeng, E.A.; Carlson, L.M.; Gutman, D.M.; Harrington, W.J., Jr.; Lee, K.P.; Boise, L.H. Proteasome inhibitors induce a terminal unfolded protein response in multiple myeloma cells. Blood 2006, 107, 4907-4916, doi:10.1182/blood-2005-08-3531.
  6. Sawamura, M.; Murakami, H.; Tsuchiya, J. Tumor necrosis factor-alpha and interleukin 4 in myeloma cell precursor differentiation. Leuk Lymphoma 1996, 21, 31-36, doi:10.3109/10428199609067576.
  7. Jurisic, V.; Colovic, M. Correlation of sera TNF-alpha with percentage of bone marrow plasma cells, LDH, beta2-microglobulin, and clinical stage in multiple myeloma. Medical oncology (Northwood, London, England) 2002, 19, 133-139, doi:10.1385/MO:19:3:133.
  8. Hong, Y.; Yu, J.; Wang, G.; Qiao, W. Association between tumor necrosis factor alpha gene polymorphisms and multiple myeloma risk: an updated meta-analysis. Hematology 2019, 24, 216-224, doi:10.1080/16078454.2018.1552341.
  9. Massague, J.; Wotton, D. Transcriptional control by the TGF-beta/Smad signaling system. The EMBO journal 2000, 19, 1745-1754, doi:10.1093/emboj/19.8.1745.
  10. Urashima, M.; Ogata, A.; Chauhan, D.; Hatziyanni, M.; Vidriales, M.B.; Dedera, D.A.; Schlossman, R.L.; Anderson, K.C. Transforming growth factor-beta1: differential effects on multiple myeloma versus normal B cells. Blood 1996, 87, 1928-1938.
  11. Kim, H.B.; Myung, S.J. Clinical implications of the Hippo-YAP pathway in multiple cancer contexts. BMB Rep 2018, 51, 119-125, doi:10.5483/bmbrep.2018.51.3.018.
  12. Cottini, F.; Hideshima, T.; Xu, C.; Sattler, M.; Dori, M.; Agnelli, L.; ten Hacken, E.; Bertilaccio, M.T.; Antonini, E.; Neri, A., et al. Rescue of Hippo coactivator YAP1 triggers DNA damage-induced apoptosis in hematological cancers. Nature medicine 2014, 20, 599-606, doi:10.1038/nm.3562.
  13. Rui, H.B.; Zheng, X.Q.; Lin, M.Y.; Yang, A.P. Sirtuin 6 promotes cell aging of myeloma cell line KM-HM_(31) by via Hippo signal pathway. European review for medical and pharmacological sciences 2018, 22, 6880-6884, doi:10.26355/eurrev_201810_16157.
  14. Shen, J.; Fu, B.; Li, Y.; Wu, Y.; Sang, H.; Zhang, H.; Lin, H.; Liu, H.; Huang, W. E3 Ubiquitin Ligase-Mediated Regulation of Osteoblast Differentiation and Bone Formation. Front Cell Dev Biol 2021, 9, 706395, doi:10.3389/fcell.2021.706395.
  15. Boyle, W.J.; Simonet, W.S.; Lacey, D.L. Osteoclast differentiation and activation. Nature 2003, 423, 337-342, doi:10.1038/nature01658.
  16. Terpos, E.; Christoulas, D.; Gavriatopoulou, M. Biology and treatment of myeloma related bone disease. Metabolism 2018, 80, 80-90, doi:10.1016/j.metabol.2017.11.012.
  17. Sugatani, T.; Alvarez, U.M.; Hruska, K.A. Activin A stimulates IkappaB-alpha/NFkappaB and RANK expression for osteoclast differentiation, but not AKT survival pathway in osteoclast precursors. Journal of cellular biochemistry 2003, 90, 59-67, doi:10.1002/jcb.10613.
  18. Terpos, E.; Kastritis, E.; Christoulas, D.; Gkotzamanidou, M.; Eleutherakis-Papaiakovou, E.; Kanellias, N.; Papatheodorou, A.; Dimopoulos, M.A. Circulating activin-A is elevated in patients with advanced multiple myeloma and correlates with extensive bone involvement and inferior survival; no alterations post-lenalidomide and dexamethasone therapy. Annals of oncology : official journal of the European Society for Medical Oncology / ESMO 2012, 23, 2681-2686, doi:10.1093/annonc/mds068.
  19. Ryoo, H.M.; Lee, M.H.; Kim, Y.J. Critical molecular switches involved in BMP-2-induced osteogenic differentiation of mesenchymal cells. Gene 2006, 366, 51-57, doi:10.1016/j.gene.2005.10.011.
  20. Standal, T.; Abildgaard, N.; Fagerli, U.M.; Stordal, B.; Hjertner, O.; Borset, M.; Sundan, A. HGF inhibits BMP-induced osteoblastogenesis: possible implications for the bone disease of multiple myeloma. Blood 2007, 109, 3024-3030, doi:10.1182/blood-2006-07-034884.
  21. Kyriazoglou, A.; Ntanasis-Stathopoulos, I.; Terpos, E.; Fotiou, D.; Kastritis, E.; Dimopoulos, M.A.; Gavriatopoulou, M. Emerging Insights Into the Role of the Hippo Pathway in Multiple Myeloma and Associated Bone Disease. Clin Lymphoma Myeloma Leuk 2020, 20, 57-62, doi:10.1016/j.clml.2019.09.620.
  22. Matsumoto, Y.; La Rose, J.; Kent, O.A.; Wagner, M.J.; Narimatsu, M.; Levy, A.D.; Omar, M.H.; Tong, J.; Krieger, J.R.; Riggs, E., et al. Reciprocal stabilization of ABL and TAZ regulates osteoblastogenesis through transcription factor RUNX2. The Journal of clinical investigation 2016, 126, 4482-4496, doi:10.1172/JCI87802.
  23. Li, B.; Shi, M.; Li, J.; Zhang, H.; Chen, B.; Chen, L.; Gao, W.; Giuliani, N.; Zhao, R.C. Elevated tumor necrosis factor-alpha suppresses TAZ expression and impairs osteogenic potential of Flk-1+ mesenchymal stem cells in patients with multiple myeloma. Stem Cells Dev 2007, 16, 921-930, doi:10.1089/scd.2007.0074.
  24. Eda, H.; Aoki, K.; Kato, S.; Okawa, Y.; Takada, K.; Tanaka, T.; Marumo, K.; Ohkawa, K. The proteasome inhibitor bortezomib inhibits FGF-2-induced reduction of TAZ levels in osteoblast-like cells. Eur J Haematol 2010, 85, 68-75, doi:10.1111/j.1600-0609.2010.01435.x.
  25. Pan, J.X.; Xiong, L.; Zhao, K.; Zeng, P.; Wang, B.; Tang, F.L.; Sun, D.; Guo, H.H.; Yang, X.; Cui, S., et al. YAP promotes osteogenesis and suppresses adipogenic differentiation by regulating beta-catenin signaling. Bone Res 2018, 6, 18, doi:10.1038/s41413-018-0018-7.
  26. Zhang, Z.Y.; Li, Y.C.; Geng, C.Y.; Wang, H.J.; Chen, W.M. Potential Relationship between Clinical Significance and Serum Exosomal miRNAs in Patients with Multiple Myeloma. BioMed research international 2019, 2019, 1575468, doi:10.1155/2019/1575468.
  27. Raimondo, S.; Urzi, O.; Conigliaro, A.; Bosco, G.L.; Parisi, S.; Carlisi, M.; Siragusa, S.; Raimondi, L.; Luca, A.; Giavaresi, G., et al. Extracellular Vesicle microRNAs Contribute to the Osteogenic Inhibition of Mesenchymal Stem Cells in Multiple Myeloma. Cancers 2020, 12, doi:10.3390/cancers12020449.
  28. Standal, T.; Seidel, C.; Hjertner, O.; Plesner, T.; Sanderson, R.D.; Waage, A.; Borset, M.; Sundan, A. Osteoprotegerin is bound, internalized, and degraded by multiple myeloma cells. Blood 2002, 100, 3002-3007, doi:10.1182/blood-2002-04-1190.

The Authors wish to thank the Reviewers for their constructive comments that led to the improvement of the current manuscript.

Round 2

Reviewer 2 Report

All of the questions were answered and the manuscript is overall improved.

Author Response

The Authors wish to thank the Reviewer.
